# Whole Genome Sequencing of the Giant Grouper (*Epinephelus lanceolatus*) and High-Throughput Screening of Putative Antimicrobial Peptide Genes

**DOI:** 10.3390/md17090503

**Published:** 2019-08-28

**Authors:** Dengdong Wang, Xiyang Chen, Xinhui Zhang, Jia Li, Yunhai Yi, Chao Bian, Qiong Shi, Haoran Lin, Shuisheng Li, Yong Zhang, Xinxin You

**Affiliations:** 1State Key Laboratory of Biocontrol, Guangdong Provincial Key Laboratory for Aquatic Economic Animals and Guangdong Provincial Engineering Technology Research Center for Healthy Breeding of Important Economic Fish, School of Life Sciences, Sun Yat-Sen University, Guangzhou 510275, China; 2Zhanjiang Bay Laboratory, Guangdong Research Center on Reproductive Control and Breeding Technology of Indigenous Valuable Fish Species, Fisheries College, Guangdong Ocean University, Zhanjiang 524088, China; 3BGI Education Center, University of Chinese Academy of Sciences, Shenzhen 518083, China; 4Shenzhen Key Lab of Marine Genomics, Guangdong Provincial Key Lab of Molecular Breeding in Marine Economic Animals, BGI Academy of Marine Sciences, BGI Marine, BGI, Shenzhen 518083, China; 5Laboratory of Aquatic Genomics, College of Life Sciences and Oceanography, Shenzhen University, Shenzhen 518060, China

**Keywords:** giant grouper, *Epinephelus lanceolatus*, genome sequencing, antimicrobial peptide, growth

## Abstract

Giant groupers, the largest grouper type in the world, are of economic importance in marine aquaculture for their rapid growth. At the same time, bacterial and viral diseases have become the main threats to the grouper industry. Here, we report a high-quality genome of a giant grouper sequenced by an Illumina HiSeq X-Ten and PacBio Bioscience Sequel platform. A total of 254 putative antimicrobial peptide (AMP) genes were identified, which can be divided into 34 classes according to the annotation of the Antimicrobial Peptides Database (APD3). Their locations in pseudochromosomes were also determined. Thrombin-, lectin-, and scolopendin-derived putative AMPs were the three largest parts. In addition, expressions of putative AMPs were measured by our transcriptome data. Two putative AMP genes (*gapdh1* and *gapdh2*) were involved in glycolysis, which had extremely high expression levels in giant grouper muscle. As it has been reported that AMPs inhibit the growth of a broad spectrum of microbes and participate in regulating innate and adaptive immune responses, genome sequencing of this study provides a comprehensive cataloging of putative AMPs of groupers, supporting antimicrobial research and aquaculture therapy. These genomic resources will be beneficial to further molecular breeding of this economically important fish.

## 1. Introduction

Groupers are coral reef fishes in the subfamily Epinephelinae of the family Serranidae (order Perciformes), which are known for their delicious taste, tender flesh, and rich nutrition [1]. As economically important fish species in marine aquaculture, groupers reached a worldwide production of 155,000 tons in 2015, with a total value of USD 630 million [2]. More specifically, mainland China is responsible for an estimated 65% of the total production [2]. There are at least 47 grouper species plus 15 grouper hybrids that have been trialed or are currently aquacultured [1]. The giant grouper, *Epinephelus lanceolatus*, is the largest grouper in the world and can grow to 2.3 m, weigh up to 400 kg [3], and it is popular for its rapid growth, reaching up to 3 kg in the first year [4]. *E. lanceolatus* itself is difficult to breed and rear; therefore, incorporating the rapid growth rate of giant groupers in the genome of hybrids has been the major focus of research on hybrid groupers [5]. Today, hybrid groupers account for a notable proportion of production. In mainland China, a commonly farmed hybrid is *E. fuscoguttatus* × *E. lanceolatus*, which is named Hulong grouper and probably accounts for more than 70% of grouper production of mainland China [2]. The molecular mechanisms underlying the superior growth of Hulong groupers have been explored based on RNA-seq, and the results showed that the upregulated expression of the upstream growth hormone and insulin-like growth factor (GH/IGF) axis related genes in the brain and liver, along with upregulated glycolytic genes as well as ryanodine receptors (RyRs) and troponins involved in the calcium signaling pathway in muscle, led to enhanced growth in the Hulong grouper [6,7].

The rapid development of the intensive culture of groupers has led to more and more severe incidences of infectious diseases, and the main impact of disease is economic loss due to reduced production [8]. It was reported that in the Asia-Pacific region, 365 diseases or disease syndrome occurrences were found in groupers, and bacterial and viral diseases constituted 40% and 26% of the problems reported, respectively [1]. Many kinds of antibiotics have been used to control bacterial diseases in grouper aquaculture; however, serious drawbacks have emerged, such as drug residues and the emergence of resistant bacterial strains [9]. Antimicrobial peptides (AMPs) are a diverse class of small cationic peptide molecules that are produced as evolutionarily ancient weapons by multicellular organisms [10]. Several kinds of AMPs from groupers, including epinecidin [11,12], hepcidin [13], defensin [14], and piscidin [15], have been cloned and studied. However, a systematic screening of AMP genes in groupers has not been reported yet.

Access to and utilization of an array of genetic resources within groupers will facilitate the analysis of AMP genes in groupers. The high-density genetic map of groupers [16] and several candidate genes and single nucleotide polymorphism (SNP) sites related to growth have been identified through quantitative trait loci (QTL) analysis [17] and genome-wide association study (GWAS) [18]. Genomic resources, especially the chromosomal-level genome assembly, would greatly help the studies in evolution, phylogeny, and biology of the groupers. In this study, we generated a high-quality genome assembly of the giant grouper by integrating the use of Illumina (San Diego, CA, USA) short reads and PacBio (Menlo Park, CA, USA) long reads. Then, the scaffolds of giant groupers were assembled based on the published high-resolution genetic map of orange-spotted grouper (*E. coioides*) [16], and they were defined as pseudochromosomes of giant groupers. Finally, glycolytic genes and putative AMP genes were identified and located in the pseudochromosomes, and the transcriptomic quantification of putative AMP genes was also determined using the available RNA-seq data [7]. These genomic recourses will be beneficial to further molecular breeding of this economically important fish.

## 2. Results

### 2.1. Genome Size Estimation

A total of 82.80 gigabases (Gb) Illumina clean data (approximately 71.07 times that of the estimated 1.165 Gb genome) and 31.15 Gb PacBio sequence data (approximately 26.74 times that of the estimated genome) were generated. A K-mer analysis was performed with clean reads of two short-insert libraries (500 and 800 bp). According to the k-mer number (20,970,039,103) and k-mer depth (18) (Figure 1), the genome size was estimated to be approximately 1.165 Gb.

### 2.2. De Novo Genome Assembly and Annotation

A total of 3,077,169 contigs were generated, and the length of contig N50 was determined to be 76,419 bp by using clean Illumina sequencing data. Subsequently, hybrid assembly involving PacBio reads that were error corrected by Illumina reads and the Illumina contigs was performed by DBG2OLC [19]. Finally, the achieved total contigs were reduced to 3207 (Table 1), which is much smaller than the assembly from Illumina data. The contig N50 and scaffold N50 of the final genome assembly were 1,469,414 and 1,505,601 bp, respectively (Table 1), reflecting the high quality of assembly. The total genome size reached up to 1.128 Gb, and the GC content was 41.4%. Our assembled genome is of high quality, as a genome completeness assessment by benchmarking universal single-copy orthologs (BUSCO) [20] proved that the assembly contained 93.1% complete gene models.

Annotation of repeat sequences was performed by de novo and homology predictions based on the RepBase database [21]. The giant grouper genome comprised 45.1% repetitive sequences with a length of 508,638,319 bp, in which 42.4% were transposable elements (TEs) with a length of 478,176,524 bp. DNA transposons (24.6% of the genome) were the most abundant type of TE, followed by long interspersed elements (LINEs, 15.8% of the genome) and long terminal repeats (LTRs, 7.4% of the genome) (Appendix A).

De novo predictions based on the repeat-masked genome, RNA-seq predictions based on transcriptomic data from our previous work [6,7], and homolog predictions generated a comprehensive and nonredundant protein-coding gene set containing 24,794 genes. Annotation completeness, assessed by BUSCO analysis, indicated that gene sequences covered 85% complete single-copy orthologs. The gene set was annotated by InterPro, KEGG, Swiss-Prot, and TrEMBL databases, and approximately 93.37% (23,149 genes) of the gene set were supported by the above-mentioned databases.

### 2.3. Pseudochromosome Construction

Based on the genetic linkage map of the orange-spotted grouper, a total of 1256 scaffolds were anchored into 24 pseudochromosomes (Chr) of the giant grouper, and a total length of 999.69 Mb was assembled, which comprised 88.62% of the assembled genome sequences and 22,206 genes (from a total of 24,794 genes). The largest pseudochromosome was Chr13 with 54.06 Mb containing 56 scaffolds, and the smallest pseudochromosome was Chr3 with 20.61 Mb containing 18 scaffolds. The average pseudochromosome length was 41.65 Mb with 52 scaffolds (Table 2 and Figure 2). Figure 2 summarizes the distribution of genes, GC content in genomic intervals of 100 kb, and interchromosomal relationships of our assembled giant grouper pseudochromosomes.

### 2.4. Identification, Transcriptomic Quantification, and Annotation of Putative Antimicrobial Peptides (AMPs)

A total of 2927 AMP sequences were collected from the online Antimicrobial Peptides Database (APD3, http://aps.unmc.edu/AP/main.php) (Appendix A), which were employed as query sequences for putative AMP identification by BLAST. A total of 254 putative AMP genes were obtained (Appendix A), which can be divided into 34 classes according to annotation of AMPs in the APD3 (Figure 4a). Each putative AMP gene was renamed by class followed by a serial number. In addition, thrombin-derived C-terminal peptides (TCPs, 64) [22,23], lectin-derived (29), and scolopendin-derived (23) were the top three classes among them, which is consistent with our previous study [24].

We also downloaded another recently published gene set of a giant grouper (PRJNA516312) from the National Center for Biotechnology Information (NCBI) and identified AMPs with the same method. We obtained 326 putative AMPs that were classified into 36 groups (Appendix A). TCPs (75), lectin-derived (46), and histone-derived (41) were the top three classes. Comparison between gene set from PRJNA516312 and present study revealed differences in the case of predicted putative AMPs. We speculated that it may be associated with differences in the annotation strategy, which resulted in divergence of the two gene sets.

Based on the transcriptomic data of brain, liver, and muscle from our previous work [6,7], transcripts per million (TPM) values of each putative AMP gene were calculated (Appendix A). TPM values reflected the transcription level of putative AMP genes. Among 254 putative AMP genes, 209, 193, and 177 putative AMP genes with TPM values were detected in brain, liver, and muscle tissues, respectively. The top 20 putative AMPs with high TPM values in each tissue are presented in Table 3.

### 2.5. Location of Putative AMP Genes and Growth-Related Genes

Out of 254 putative AMPs, 228 were mapped to the 24 assembled pseudochromosomes, with an average number of 9 per chromosome (Figure 3). Chr10 and Chr15 had the highest hits of 22 and 19 genes, respectively. The chromosomes with the lowest counts were Chr20 and Chr3, both with 3 genes. Subsequently, putative AMPs were also confirmed by KEGG enrichment analysis, in which 167 genes were clustered into 35 KEGG items. The representative entries included the immune system (53 genes), signaling molecules and interactions (48 genes), signal transduction (46 genes), and cancers: overview (37 genes) (Figure 4b). Among them, glycogen synthesis may be related to the superior growth of groupers [6]. Thus, in this study, we found that 24 glycolytic- and Ca^2+^-regulating putative AMP genes were located in 18 chromosomes. We found that *gapdh2*, *eno2,* and *tpi1a* were located in Chr22 (Table 4 and Figure 3).

Interestingly, we found that 2 genes (*gapdh1* and *gapdh2*) involved in glycolysis were precursors of predicted AMPs (Figure 3). The expression level of *gapdh1* in the giant grouper muscle was extremely high (Table 5), which has a high identity (96.88%) and query alignment ratio (100%) with the skipjack tuna GAPDH-related antimicrobial peptide (SJGAP) [25]. While *gapdh2* matched to yellowfin tuna glyceraldehyde-3-phosphate dehydrogenase-related antimicrobial peptide (YFGAP) [25], with 87.50% identity and 100.00% query alignment ratio. SJGAP and YFGAP are AMPs from the skin of skipjack tuna (*Katsuwonus pelamis*) and yellowfin tuna (*Thunnus albacares*), respectively, and both have potent antimicrobial activities [25,26]. To investigate the alignments of SJGAP and YFGAP in *gapdh*, we performed multiple sequence alignments of *gapdh1* (Figure 5a) and *gapdh2* (Figure 5b) from zebrafish, yellowfin tuna, and giant groupers. As shown in Figure 5, *gapdh1* of zebrafish, yellowfin tuna, and giant groupers showed higher similarity with YFGAP and SJGAP than *gapdh2*, suggesting that *gapdh1* are more likely to play a role in the antimicrobial process in these fishes. Gene structures of *gapdh1* and *gapdh2* are also exhibited in Figure 5.

## 3. Discussion

Along with the publication of genomes of model organisms and species with specific evolutionary characteristics, an increasing number of important crops and economic animals have been sequenced, as in the case of rice [27], wheat [28], barley [29], maize [30,31], soybeans [32], and cotton [33] as well as cows [34], pigs [35,36], sheep [37], goats [38,39], cod [40], sea basses [41,42], mudskippers [43], salmonids [44,45], carps [46,47], and tongue sole [48]. In this study, we sequenced the genome of giant groupers with the purpose of systematically gaining its genetic information and providing opportunities to accelerate breeding improvement.

Most of the genomes of economic crops or animals have focused on growth traits. While some fish genome projects found immune genes that were lost (elephant shark [49], Atlantic cod [40]) or expanded (large yellow croaker [50]), we performed this project to screen putative AMPs with an attempt to explore immune resources for bacterial and viral disease therapy. Especially, some AMPs have been widely used in agriculture as potential alternatives to antibiotics [51]. This work may help those who make efforts to develop drugs for groupers and reduce the usage of antibiotics and other chemically poisoning drugs.

A total of 254 putative AMP genes of the giant grouper were classified into 34 groups in the present study. Among them, thrombin (64 AMPs), lectin (29 AMPs) and scolopendin (23 AMPs) were the top three groups. Thrombin was also the largest grouper in our previous works, including the blue tilapia (*Oreochromis aureus*), Nile tilapia (*Oreochromis niloticus*) [52], blue-spotted mudskipper (*Boleophthalmus pectinirostris*), and giant-fin mudskipper (*Periophthalmus magnuspinnatus*) [24]. Lectin also accounted for a major part in the above-mentioned four fishes. It seems that thrombin may play a vital role in fish.

It has been reported that several kinds of AMPs, including epinecidin [11,12], hepcidin [13], defensin [14], and piscidin [15], have been cloned and studied from groupers. We identified EC-hepcidin1 (query ID 1701 in Appendix A), a hepcidin AMP derived from the liver and stomach of orange-spotted grouper [13], from the giant grouper gene set. Enap-1 (query ID 294 in Appendix A), a defensin from horse (*Equus caballus*) [53], was also identified in giant groupers. However, the other two AMPs that have been reported in groupers were not found in our annotated gene set.

Cumulative evidence is showing that AMPs not only inhibit the growth of a broad spectrum of microbes through membrane disruption [10], but they also participate in regulating innate and adaptive immune responses [54,55]. High expression levels of the *gapdh1* gene in the liver and muscle of giant groupers may imply its active glycolysis activity along with antimicrobial activity. Previous studies have reported that strong antibacterial activities against both Gram-positive and Gram-negative bacteria of N-terminal segments of this protein were shown in tuna fish [25,26]. Antifungal efficacies of GAPDH-derived peptide were also demonstrated in many studies [56,57]. However, GAPDH has not been shown to be cleaved in groupers, and it is possible that cleavage might be tissue specific. Even if it has a high level in muscle, it would be predicted to be at full length and enzymatically active for glycolysis only, whereas it might be cleaved to AMPs in skin or other tissues. Antimicrobial functions of this peptide (if produced) are worthy of further investigation in giant groupers.

Moreover, the great majority of giant groupers are born to be female, barely male, and some, but mostly one of them in the population, would change its gender to male after the first or second maturation (all females have the ability to change sex). This kind of protogynous hermaphrodite is mostly shared by groupers [58]. Materials in this work could provide opportunities to explore the mechanisms of sex change in groupers.

## 4. Materials and Methods

### 4.1. Sample Preparation and Sequencing

Genomic DNA was extracted from the muscle of a wild giant grouper cultured in the Guangdong Marine Fishery Experimental Center, Huizhou, China. Six libraries, including three short-insert libraries (270, 500, and 800 bp) and three long-insert libraries (2, 5, and 10 kb), were constructed for sequencing by an Illumina HiSeq X-Ten platform (Illumina, San Diego, CA, USA), except for the 800 bp insert-size library, which was sequenced by an Illumina HiSeq 2500 platform (Illumina, San Diego, CA, USA). For high-quality assembly, we also constructed a 20 kb insert library for sequencing with the PacBio Bioscience Sequel platform (Pacific Biosciences, Menlo Park, CA, USA). All experiments were carried out according to the guidelines of the Animal Ethics Committee and were approved by the Institutional Review Board on Bioethics and Biosafety of BGI (No. FT14015).

All sequenced data from *E. lanceolatus* are available in the NCBI database at BioProject ID, PRJNA533524. All Illumina reads are available under accession numbers SRR8926032, SRR8926031, SRR8926030, SRR8926029, SRR8926035, SRR8926034, and SRR8926033 in the NCBI database.

### 4.2. Genome Assembly

The Illumina raw sequences with adapter contamination, low quality, and replicated PCR were filtered out by a SOAPfilter (version 2.2, BGI-Shenzhen, Shenzhen, China). PacBio raw data were corrected by Illumina clean reads through LoRDEC (version 0.4.1, http://www.atgc-montpellier.fr/lordec/) [59]. The Illumina clean reads were assembled by Platanus (version 1.2.4, Tokyo Institute of Technology, Tokyo, Japan) [60] to construct contigs. Subsequently, contigs were aligned to PacBio reads by DBG2OLC [19] to construct consensus contigs. Finally, the assembled genome was polished by Pilon (version 1.22, Broad Institute of MIT and Harvard, Cambridge, MA, USA) [61] with short-insert library reads.

### 4.3. Pseudochromosome Construction

SNP-containing reads in the genetic linkage map from the orange-spotted grouper [16] were mapped to the giant grouper assembled genome sequence, and only the matching reads were selected. Linkage groups (LGs) of the giant grouper were assigned using JoinMap4.1 software (Kyazma, Wageningen, Netherlands) [62]. Subsequently, a genetic linkage map was constructed. Single nucleotide polymorphisms in the genetic linkage map of the giant grouper were used for assembling the pseudochromosomes. To increase the accuracy of pseudochromosome assembly, we chose at least two SNPs in each scaffold [63]. Based on genetic distances between SNP markers, we determined the position and orientation of each scaffold and anchored these scaffolds to construct pseudochromosomes.

### 4.4. Repeat Annotation

Repeat elements were predicted by de novo and homology methods. De novo predictions were performed by LTR_FINDER (version 1.0.6, Fudan University, Shanghai, China) [64] and RepeatModeler (version 1.08, http://www.repeatmasker.org/RepeatModeler/) [65]. The merged repeat library was aligned to the assembled genome sequence by RepeatMasker (version 4.06, Institute for Systems Biology, Seattle, WA, USA) to produce repeat elements [65]. The homology prediction based on RepBase (version 21.01, Genetic Information Research Institute, Sunnyvale, CA, USA) was performed by RepeatMasker and RepeatProteinMask (version 4.06, Institute for Systems Biology, Seattle, WA, USA) [65]. Subsequently, nonredundant repeat elements were obtained by integrating de novo and homology data.

### 4.5. Gene Annotation

We applied three different strategies to predict the protein-coding genes. For the de novo prediction strategy, AUGUSTUS (version 2.5, Institute of Microbiology and Genetics, University of Göttingen, Göttingen, Germany) [66] and GENSCAN (version 1.0, Stanford University, Stanford, CA, USA) [67] were employed to predict genes from the repeat-masked genome. The second strategy was homology-based annotation. Protein sequences of three-spined stickleback (*Gasterosteus aculeatus*), spotted gar (*Lepisosteus oculatus*), Nile tilapia (*Oreochromis niloticus*), medaka (*Oryzias latipes*), Japanese puffer (*Takifugu rubripes*), spotted green pufferfish (*Tetraodon nigroviridis*), platyfish (*Xiphophorus maculatus*), and zebrafish (*Danio rerio*) were downloaded from Ensembl database (release version 90, https://asia.ensembl.org/index.html). Protein sequences of Asian seabass (*Lates calcarifer*) were downloaded from http://seabass.sanbi.ac.za. Downloaded protein sequences were aligned to the assembled genome of the giant grouper by TBLASTN (e-value: 1e−5) [68]. GeneWise (version 2.2.0, The European Bioinformatics Institute, Cambridge, UK) [69] was used to predict the gene structure of each BLAST hit. For the third strategy, transcriptome-based prediction, raw data were downloaded from the National Center for Biotechnology Information (NCBI), including liver, brain, and muscle transcriptomic data. Subsequently, we employed TopHat (version 2.1.1, Johns Hopkins University, Baltimore, MD, USA) [70] and cufflinks (version 2.2.1, http://cufflinks.cbcb.umd.edu/) [71] with raw reads to predict protein-coding genes. As a result, integrated and nonredundant gene sets were obtained by GLEAN [72] with the above-mentioned three results.

The predicted gene sets were aligned to InterPro [73], KEGG [74], TrEMBL, and Swiss-Prot [75] databases to accomplish functional annotation.

### 4.6. Identification and Transcriptomic Quantification of AMPs

A total of 2927 AMP sequences that have been reported to exhibit antimicrobial activity were collected from the APD3 database as a query sequence (Appendix A). An index database of annotated gene sets was built for alignment by makeblastdb command. Collected active AMPs sequences were aligned to gene set sequences to identify potential AMPs based on sequence similarity by TBLASTN (e-value: 1e−5). Alignment hits were dealt with by in-house scripts. Those hits with a query alignment ratio less than 0.5 were filtered out, and redundant data were also removed. To calculate the TPM value of putative AMP genes, we performed referring sequence-based transcript quantification. Raw reads were filtered by SOAPnuke filter tools (version 1.5.6, BGI-Shenzhen, Shenzhen, China) [76]. Clean reads were mapped to the assembled genome by HISAT2 (version 2.0.4, https://github.com/DaehwanKimLab/hisat2) [77]. Subsequently, TPM values of each transcriptome were calculated by RSEM (version 1.2.12, https://deweylab.github.io/RSEM/) [78].

## 5. Conclusions

We report a high-quality genome of the giant grouper. The assembly reached up to 1.128 Gb, accounting for 96.8% of the estimated genome size. A total of 24,794 protein-coding genes were annotated through de novo prediction, transcriptomic data, and homolog prediction. Then, 254 putative AMP genes were identified, located in pseudochromosomes, and expressions were measured. Two putative AMP genes were connected to glycolysis, of which *gapdh1* was highly expressed in muscle. Genome sequencing let us identify AMPs systematically in groupers so as to support antimicrobial research and possibly provide suggestions for therapy. These genomics resources will be beneficial for further molecular breeding of this economically important fish. This work shall aid in the effort against infectious diseases in the giant grouper industry.

## Figures and Tables

**Figure 1 marinedrugs-17-00503-f001:**
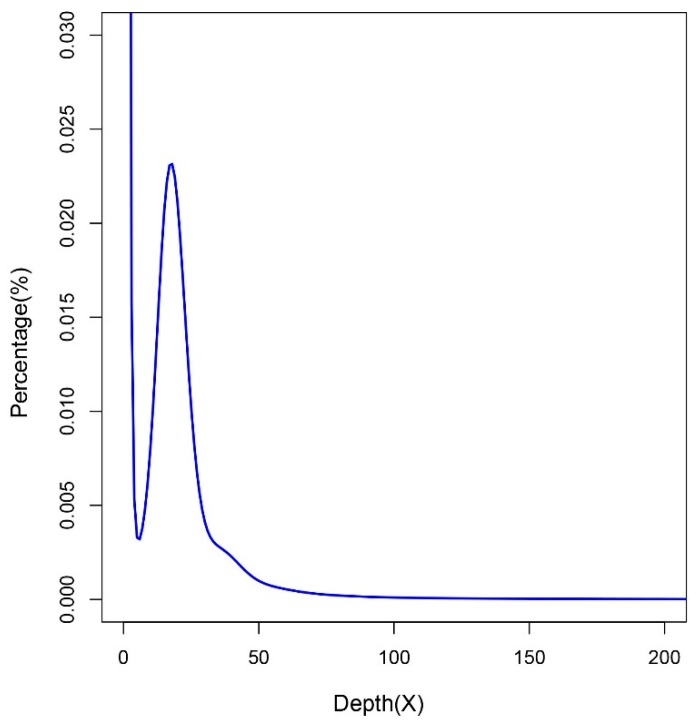
K-mer distribution of the giant grouper with a k-mer size of 17. The *x*-axis and *y*-axis are the sequencing depth and percentage of unique 17-mers, respectively.

**Figure 2 marinedrugs-17-00503-f002:**
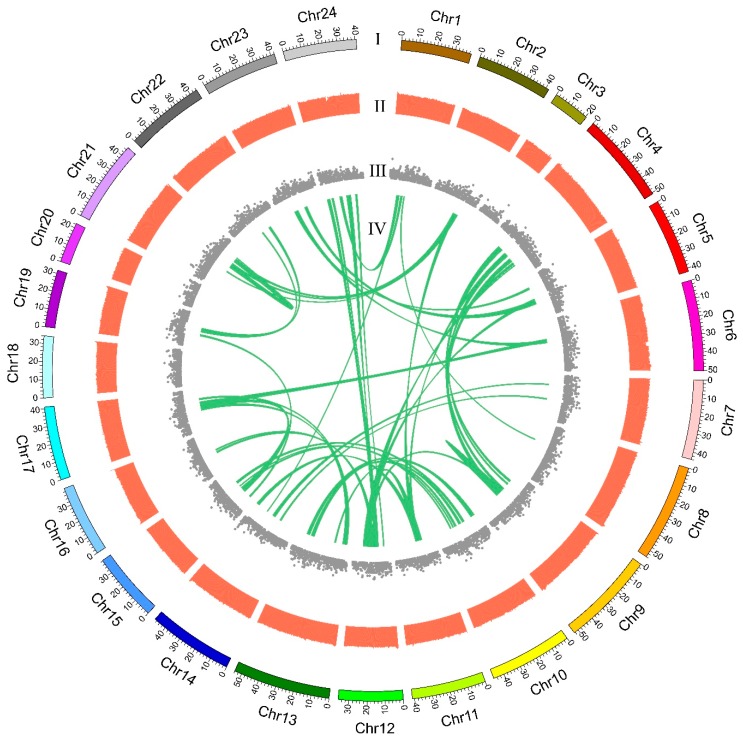
Circos atlas representation of pseudochromosome information. (**I**) The length of each pseudochromosome. (**II**) GC content of 100-kb genomic intervals (GC content from 0.25 to 0.51). (**III**) Density of gene distribution in each 100-kb genomic interval. (**IV**) Schematic presentation of major interchromosomal relationships in the giant grouper genome, which represents the collinearity of genes between two chromosomes.

**Figure 3 marinedrugs-17-00503-f003:**
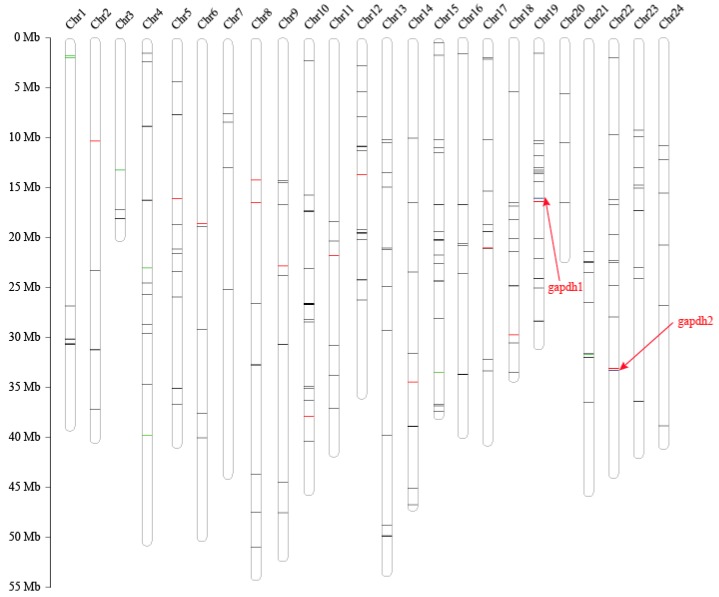
Pseudochromosome lengths, genes involved in AMPs and glycolysis, and Ca^2+^-regulating genes of the giant grouper. NOTE: Black bars represent AMPs; red bars represent genes involved glycolysis; green bars represent genes involved Ca^2+^ regulation; and blue bars represent the gapdhs, which are both AMPs and glycolysis.

**Figure 4 marinedrugs-17-00503-f004:**
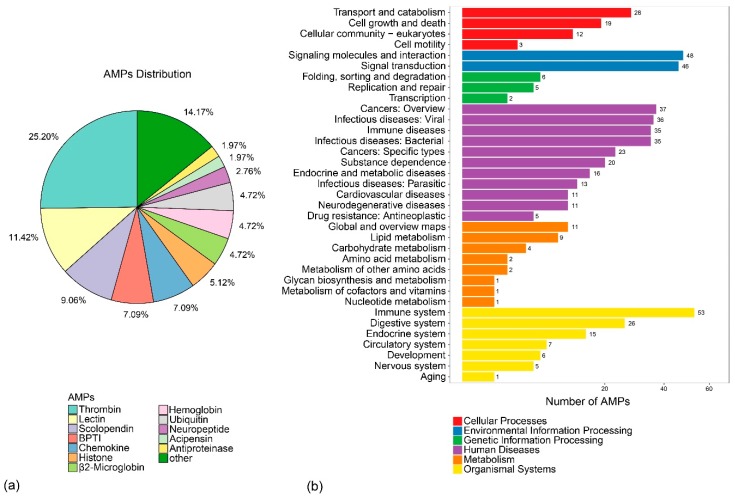
(**a**) AMP distribution and (**b**) KEGG metabolic pathway annotation of AMPs.

**Figure 5 marinedrugs-17-00503-f005:**
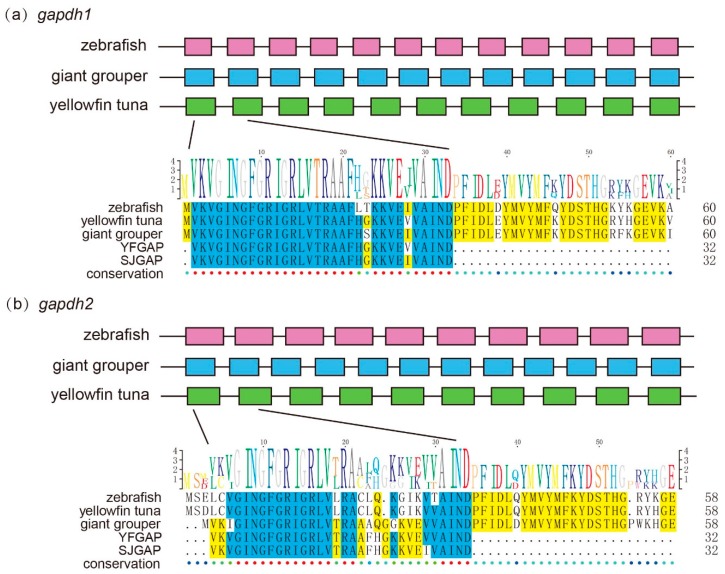
Structure of the 2 growth-related putative AMP genes (*gapdh1* and *gapdh2*) in zebrafish, giant groupers, and yellowfin tuna. The pink, blue, and green boxes represent coding sequence (CDS) of genes in zebrafish, giant groupers, and yellowfin tuna, respectively. Multiple sequence alignments of partial *gapdh1* (**a**) and *gapdh2* (**b**) that match to YFGAP and SJGAP from zebrafish, yellowfin tuna, and giant grouper were displayed. Blue and yellow marks represent >80% and >50% identity, respectively.

**Table 1 marinedrugs-17-00503-t001:** Genome assembly statistics of the giant grouper.

Criteria	Contig	Scaffold
Number	3207	3187
Total length (bp)	1,128,030,970	1,128,030,990
Longest (bp)	9,533,321	9,533,321
N50 (bp)	1,469,414	1,505,601
N90 (bp)	209,611	210,944
>2 kb	3182	3162

**Table 2 marinedrugs-17-00503-t002:** Characteristics of pseudochromosomes of *Epinephelus lanceolatus.*

Chr	Length (Mb)	Number of Genes	Number of Scaffolds
1	39.05	917	53
2	41.84	899	54
3	20.61	374	18
4	50.74	1198	78
5	41.95	912	53
6	50.39	978	72
7	44.01	1062	55
8	54.01	1051	56
9	52.56	1127	70
10	45.93	1209	49
11	41.12	820	41
12	35.95	918	45
13	54.06	1359	56
14	47.27	947	60
15	38.31	894	35
16	39.82	728	57
17	40.89	858	47
18	34.17	675	58
19	31.81	898	24
20	22.54	491	24
21	45.46	1007	68
22	44.25	944	80
23	42.07	1035	43
24	40.81	905	60
total	999.69	22,206	1256

**Table 3 marinedrugs-17-00503-t003:** The top 20 transcripts per million (TPM) rankings of antimicrobial peptides (AMPs) or putative AMP precursors in each of the three transcriptomic datasets.

Ranking	Muscle	Liver	Brain
1	GAPDH1 ^1^ (49,379.59)	Hemoglobin1 (24,013.91)	Hemoglobin1 (11,531.97)
2	GAPDH2 (5017.39)	Hemoglobin12 (23,219.52)	Hemoglobin12 (7262.71)
3	Hemoglobin12 (2440.09)	sOT2 ^2^ (14,410.88)	sOT2 (1416.06)
4	Hemoglobin1 (2295.59)	Antiproteinase1 (11,271.67)	β2-Microglobin1 (1136.57)
5	Ap-s ^3^ (339.63)	Antiproteinase5 (7898.22)	Neuropeptide5 (858.07)
6	β2-Microglobin1 (258.26)	Antiproteinase2 (6566.73)	BPTI4 ^4^ (654.21)
7	β2-Microglobin4 (137.71)	Thrombin1 (5759.76)	Neuropeptide6 (631.31)
8	Ubiquicidin (86.92)	GAPDH1 (5378.38)	Saposin2 (572.89)
9	BPTI16 (85.56)	BPTI12 (4143.13)	Lectin12 (535.93)
10	Saposin2 (81.02)	Thrombin23 (2039.84)	Synuclein (350.56)
11	BPTI7 (57.05)	Antiproteinase3 (1806.51)	β2-Microglobin4 (340.17)
12	Lectin25 (43.21)	Thrombin29 (1544.98)	Amyloid2 (250.76)
13	Thrombin53 (40.79)	Thrombin22 (1313.59)	Amyloid1 (249.58)
14	Thrombin31 (38.01)	Thrombin46 (1201.45)	Lysozyme2 (208.82)
15	BPTI15 (32.09)	Thrombin6 (1039.43)	Ubiquicidin (191.88)
16	CcAMP ^5^ (30.21)	β2-Microglobin1 (662.14)	Thymosin2 (177.76)
17	BPTI14 (27.33)	Thrombin64 (621.69)	Thrombin45 (171.27)
18	Lectin3 (23.87)	Thrombin42 (590.54)	Ubiquitin10 (119.27)
19	Ubiquitin1 (23.61)	Thrombin47 (554.85)	LEAP-2_2 ^6^ (116.44)
20	Ubiquitin5 (21.93)	Thrombin18 (473.18)	Lectin19 (103.89)

^1^ glyceraldehyde 3-phosphate dehydrogenase; ^2^ an AMP derived from *Pelodiscus sinensis*; ^3^ an AMP purified from *Argopecten purpuratus*; ^4^ bovine pancreatic trypsin inhibitor; ^5^ an AMP from *Coridius chinensis*; ^6^ liver-expressed antimicrobial peptide 2. NOTE: TPM values of putative AMPs are exhibited in parentheses.

**Table 4 marinedrugs-17-00503-t004:** Location of glycolytic- and Ca^2+^-regulating genes in the giant grouper pseudochromosome.

Gene Name	Chr	Gene ID	Function Type
*tni-fast* ^1^	Chr1	longdun_GLEAN_10010987	Ca^2+^ regulating
*tnt-skeletal* ^2^	Chr1	longdun_GLEAN_10010985	Ca^2+^ regulating
*pgk* ^3^	Chr2	longdun_GLEAN_10005384	Glycolytic
*tni-slow*	Chr3	longdun_GLEAN_10021266	Ca^2+^ regulating
*tnc* ^4^	Chr4	longdun_GLEAN_10018260	Ca^2+^ regulating
*tnt-cardiac*	Chr4	longdun_GLEAN_10017734	Ca^2+^ regulating
*pgam2* ^5^	Chr5	longdun_GLEAN_10022627	Glycolytic
*pgm2* ^6^	Chr6	longdun_GLEAN_10022325	Glycolytic
*gPi* ^7^	Chr8	longdun_GLEAN_10012880	Glycolytic
*pyk* ^8^	Chr8	longdun_GLEAN_10018335	Glycolytic
*pfk-muscle* ^9^	Chr9	longdun_GLEAN_10019679	Glycolytic
*ald* ^10^	Chr10	longdun_GLEAN_10019814	Glycolytic
*pfk-liver*	Chr11	longdun_GLEAN_10018530	Glycolytic
*pgm1*	Chr12	longdun_GLEAN_10020822	Glycolytic
*pgm3*	Chr14	longdun_GLEAN_10008616	Glycolytic
*ryr2* ^11^	Chr15	longdun_GLEAN_10008973	Ca^2+^ regulating
*pgam1a*	Chr17	longdun_GLEAN_10012515	Glycolytic
*eno1* ^12^	Chr18	longdun_GLEAN_10002289	Glycolytic
*gapdh1* ^13^	Chr19	longdun_GLEAN_10017174	Glycolytic
*tpi1b* ^14^	Chr19	longdun_GLEAN_10017191	Glycolytic
*ryr1*	Chr21	longdun_GLEAN_10010008	Ca^2+^ regulating
*gapdh2*	Chr22	longdun_GLEAN_10014462	Glycolytic
*eno2*	Chr22	longdun_GLEAN_10014466	Glycolytic
*tpi1a*	Chr22	longdun_GLEAN_10014467	Glycolytic

^1^ troponin I; ^2^ troponin T; ^3^ phosphoglycerate kinase; ^4^ troponin C; ^5^ phosphoglycerate mutase; ^6^ phosphoglucomutase; ^7^ phosphoglucose isomerase; ^8^ pyruvate kinase; ^9^ phosphofructokinases; ^10^ fructose-bisphosphate aldolase; ^11^ ryanodine receptor; ^12^ enolases; ^13^ glyceraldehyde phosphate dehydrogenase; and ^14^ triosephosphate isomerase.

**Table 5 marinedrugs-17-00503-t005:** Congruent relationship and transcription levels of the 2 growth-related putative AMP genes.

Putative AMP Gene Name	Gene ID	Query AMP (AMP ID in APD3 Database)	TPM		
			Brain	Liver	Muscle
*gapdh1*	longdun_GLEAN_10017174	Skipjack tuna GAPDH-related antimicrobial peptide (SJGAP) (2680)	4.77	5378.38	49379.59
*gapdh2*	longdun_GLEAN_10014462	Yellowfin tuna glyceraldehyde-3-phosphate dehydrogenase-related antimicrobial peptide (YFGAP) (2012)	12.19	0.17	5017.39

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
