# Peer review of "Whole Genome Sequencing of the Giant Grouper (Epinephelus lanceolatus) and High-Throughput Screening of Putative Antimicrobial Peptide Genes"

_marinedrugs, 2019, doi:10.3390/md17090503_

Round 1
Reviewer 1 Report
In this article, authors reported high-quality genome of the giant grouper. It is an important aquaculture species which very popular in Asian countries because of fast growth during the first year of life. The quality of genome assembly and methods which were used during assembling don’t give rise to any substantive questions. The main problem of this article is totally absent of any idea. I think it is the latest fashion to publish genome with the prediction of antimicrobial peptides because I suppose that this manuscript should be ideologically improved before the publishing.
General comments:
There is another recently published (2019/05/13) high-quality genomic sequence of Epinephelus lanceolatus (PRJNA516312). I think you have to cite it and try to compare your findings in the case of predicted antimicrobial peptides. Moreover, it should interesting to compare the presence and diversity of antimicrobial peptides with other species from Serranidae family (e.g. Epinephelus moara – PRJNA543191, Hypoplectrus puella - PRJEB27858). Because you tried to analyze transposable elements in giant grouper I totally recommend to create comprehensive plots for activity this part of the genome in Serranidae family (the good example of this analysis is presented here: PMID: 29715568). Using RNA-Seq data authors described 20 top ranking AMPs in liver, muscle, and brain (Table 3). I got a little bit embarrassed because in this table there a lot of potentially “blood” AMPs. I am not sure that these AMPs which were found are related to named tissues. Also, it looks strange that authors don’t discuss their findings with comparing recently published data (e.g. it is well known that hepcidin is very common in the fish liver (PMID: 30659854, PMID: 30648626). It should be interesting to conduct analysis previously described in this article (PMID: 29713520) to show the patterns of AMP presence through the genome are they located randomly or not.
Minor comments:
Line 58: The meaning of abbreviations is needed Line 96: Transfer this sentence to the Material and methods section.
Author Response
In this article, authors reported high-quality genome of the giant grouper. It is an important aquaculture species which very popular in Asian countries because of fast growth during the first year of life. The quality of genome assembly and methods which were used during assembling don’t give rise to any substantive questions. The main problem of this article is totally absent of any idea. I think it is the latest fashion to publish genome with the prediction of antimicrobial peptides because I suppose that this manuscript should be ideologically improved before the publishing.
Answer: Thanks for your comments. In this study, we reported a high-quality genome of giant grouper sequenced by Illumina HiSeq X-Ten and PacBio Bioscience Sequel platform. As important aquaculture species, a systematic screening of AMP genes could support anti-microbial research and aquaculture therapy of groupers. Thereby, a total of 254 putative antimicrobial peptide (AMP) genes were identified from the annotated gene set. Two putative AMP genes (gapdh1 and gapdh2) were involved in glycolysis, which had extremely high expression level in giant grouper muscle. Then, those glycolytic and Ca2+ regulation related genes were located and were also presented in table 4, which were candidate genes for illustrating the trait of fast growth of giant grouper.
To further determined the potential function of these two putative AMP genes (gapdh1 and gapdh2), multiple sequence alignment of gapdh1 and gapdh2 (see the revised Figure 5) from zebrafish, yellowfin tuna and giant grouper was performed, and we found gapdh1 of zebrafish, yellowfin tuna and giant grouper showed higher similarity with YFGAP and SJGAP than gapdh2, and the previous studies (PMID: 22771964; PMID: 24412436 ) showed that skipjack tuna GAPDH-related antimicrobial peptide (SJGAP) and yellowfin tuna glyceraldehyde-3-phosphate dehydrogenase-related antimicrobial peptide (YFGAP) are AMPs from the skin of skipjack tuna (Katsuwonus pelamis) and yellowfin tuna (Thunnus albacares) respectively, both with potent antimicrobial activity. We suggested that gapdh1 are more likely to play a role in antimicrobial process in these fishes. These statements were added in the lines 202-211 of revised manuscript.
General comments:
There is another recently published (2019/05/13) high-quality genomic sequence of Epinephelus lanceolatus (PRJNA516312). I think you have to cite it and try to compare your findings in the case of predicted antimicrobial peptides.
Answer: Thanks for your suggestion. We downloaded the gene set of another published genome (PRJNA516312) from NCBI and identified AMPs with the same method. We obtained 326 putative AMPs that were classified into 36 groups (Table S4). TCPs (75), lectin-derived (46) and histone-derived (41) were the top three classes. Comparison between gene set of another genome assembly (PRJNA516312) and present study revealed difference in the case of predicted putative AMPs. We speculated that it may be associated with differences in annotation strategy, which resulted to divergence of the two gene sets.
Added analysis please refer to lines 154-159 and new supplementary table (Table S4).
Moreover, it should interesting to compare the presence and diversity of antimicrobial peptides with other species from Serranidae family (e.g. Epinephelus moara – PRJNA543191, Hypoplectrus puella - PRJEB27858).
Answer: Thanks for your suggestion. We failed to obtain the gene set of Epinephelus moara and Hypoplectrus puella, because the annotation of these two genome assemblies has not been completed. Thereby, we can not carry out the comparative analysis.
Because you tried to analyze transposable elements in giant grouper I totally recommend to create comprehensive plots for activity this part of the genome in Serranidae family (the good example of this analysis is presented here: PMID: 29715568).
Answer: Thanks for your suggestion. Analyzing the TEs is the routine workflow of genome annotation. The TEs play unique and diverse roles as presented in your suggested reference. As your recommendation, it would be helpful and interesting to analyze TEs among Serranidae genomes even all available fish genomes, however, the systematic analysis of TEs would be probably out of the scope of this journal. We will focus on this in the future’s work.
Using RNA-Seq data authors described 20 top ranking AMPs in liver, muscle, and brain (Table 3). I got a little bit embarrassed because in this table there a lot of potentially “blood” AMPs. I am not sure that these AMPs which were found are related to named tissues. Also, it looks strange that authors don’t discuss their findings with comparing recently published data (e.g. it is well known that hepcidin is very common in the fish liver (PMID: 30659854, PMID: 30648626).
Answer: Thanks for your comments. It is true that fragments of hemoglobin are AMPs (PMID: 27919228, PMID: 18538841), which expressed in liver, muscle and brain at high levels, consistent with our previous study (PMID: 29165344). In this study, we performed a high-throughput identification of AMPs in the giant grouper and found there are 254 potential AMPs. GAPDH raised our interest by high expression levels and growth-related characteristics. Hepcidin was found to have a low TPM value in liver (0.33) while not detected in muscle and brain in this study, which may be difficult to compare with other data while their sequences have high similarity. This big-data-based analysis may not do well in a specific AMP family.
It should be interesting to conduct analysis previously described in this article (PMID: 29713520) to show the patterns of AMP presence through the genome are they located randomly or not.
Answer: Thanks for your comments. As the Fig. 3 showed, the 228 putative AMPs were scattered located on 24 assembled pseudo-chromosomes. We also marked the position of AMPs ranked by transcripts per million (see the attached file “Table_S5”), and we didn’t find notable co-expressed clusters as described in that article (PMID: 29713520). Therefor, the patterns of AMP presence through the genome are they located randomly.
Minor comments:
Line 58: The meaning of abbreviations is needed
Answer: Yes, the full name was added in the revised manuscript.
Line 96: Transfer this sentence to the Material and methods section.
Answer: Yes, this sentence had been transferred to the Material and methods section.
Reviewer 2 Report
The manuscript by Wang and colleagues describes a whole genome sequencing project for the economically important giant grouper fish, together with identification of putative antimicrobial peptide genes (and their relative transcript abundance in three issues). Identification of AMPs is sought to exploit in protection versus pathogens. Overall the manuscript is straightforward in scope, but much of the manuscript is focused on the genome sequence generation and assembly itself (which are not trivial), and the focus is confused by the blending of information relating to AMPs as well as glycolysis and growth. I appreciate that there is some overlap with the identification of GAPDH in both classes.
Specific Comments
I suggest that the authors qualify some of the identifications as “putative” antimicrobial peptide genes (e.g. in the title). As indicated in the Introduction, L69, AMPs are generally considered to be small cationic peptides but most of the proteins identified in this study are larger (Thrombin homologs, GAPDH etc). This is confounded by the fact that some AMPs are generated by post-translational processing (presumably) from larger precursors. The fact that a small peptide from Tuna GAPDH has AMP activity, does not mean that an N-terminally derived peptide is even generated in Grouper, even if the sequences are closely related. In addition, the authors use E-05 as a cut off for query searches so some of the hits are likely to be only distantly related and therefore may not have the activity ascribed to one protein. For these reasons, the term putative AMP should be used. Section 2.3 The authors should aid the reader by defining what is meant as a pseudo-chromosome in the context of their study. Figure 2. There is no scale for ring II, and all red bars seem to be the same height (at this resolution). In addition, the green linkages in ring IV should be further explained. What is a major “inter-chromosomal relationship” that is being shown? Sequence match? How much similarity? length? Cut off? L140. The authors should comment on whether it is common or unusual to have 64 thrombin like genes, 29 lectin like genes in a fish genome. Without this information it is difficult to put into biological perspective. Although the authors report some of the highly expressed AMPs (or putative AMPs), it would be informative to include in the text whether the five previously found grouper AMPs listed in the Introduction, were found in the giant grouper genome. Table 4 focus es on glycolytic and Ca2+ regulation. This detracts from the theme of the paper (and would appear to be out of scope for the journal). Figure 5. Why is zebrafish shown when the published reference included in the text relates to tuna? If the zebrafish is known to have a N-terminal peptide produced, that reference should be included.
Minor comments (mostly language suggestions)
L27 At the same time L28 the grouper industry L29 HiSeq and Pacific Biosciences (also L224) L31 of the APD3 database. L31. Use another verb rather than implemented. L37. I’m not sure that it is a “systematic exploration” but perhaps a “comprehensive cataloging” L45 Please provide a reference for this fact. L47 More specifically mainland China is responsible…. L50 and can grow….weighs…. L53 hybrids L56 and probably accounts for more than 70% of total…. L48 and L56. There may be a discrepancy (or further clarification is needed). If China produces 65% of Grouper production, how can Hulong account for 70% of all grouper (assuming this is all in China)? L66 respectively L67 drawbacks emerged , such as L73. This paragraph needs better linkage/transition to previous paragraph L75 analysis L76 assembly would be a great help for the L79 PacBio L83 These genomic resources (check spelling) L89 500 bp and 800 bp (not kb, insert spaces) L98 that were error L101 Gb L102 genome completeness L104. The authors should clarify. Does this mean that 93% of the genes expected, were present in the final assembly? The authors should mention in the text that the final assembly had 3207 contigs (this is more important than knowing that Illumina had 3 million (L97) L107 and L250 RepBase or Repbase? Also, please provide a reference. L108 Define ‘TE” on first use. L114. Here, 85% or othologs are reported to be present—but see Point 24 above where 93% gene models are present. This is unclear. L115 The gene set L120 and throughout; both Chr and chr are used as an abbreviation. Please use one style throughout (and in Figures/Tables) L121 (from a total of …) L140 and elsewhere. More discussion should be included. It is not that thrombin is an AMP (it is too big), but in certain species, proteolytic fragments of thrombin have been shown to have AMP activity. This should be included together with a reference to help readers less familiar with the concept. L145 are presented in Table 3 Table 3 should have “AMPs or putative AMP precursors” in the title. Many (most) of these are large proteins, not small AMPs. The footnote should indicate what the numbers in parentheses refer to. L159, L171 Table 4 title. Ca2+ regulated genes. If Table 4 is retained (I feel it should not), it should be clearly indicated which are glycolytic genes and which are calcium regulation genes Fig 3. Which colour are the Ca-regulated genes? Only black (AMP) and red (glycolysis) are labelled. GapDH should be both colours. L176. This sentence reads like it is a new discovery—but has been known for decades. This is the expected role for GAPDH—the AMP role is more novel. L179 is exhibited in Figure 5 L176-181. And L207 especially; The authors should comment on the fact that GAPDH has not been shown to be cleaved in Grouper and it is possible that cleavage might be tissue specific. So even if high levels in muscle, it would be predicted to be full length and enzymatically active for glycolysis whereas in skin or other tissues it might be cleaved to AMPs. L199 we performed ..to screen for L201 Please provide a reference for this fact L209 of N-terminal segments of this protein was shown L211 of this peptide (if produced) is worthy… L232 data were corrected L234 contigs were aligned L241 a genetic L243 pseudo-chromosome assembly L272 have been reported to exhibit antimicrobial L275 identify potential AMPs L277 data were L283 Gb L284 accounting for 96.8% L286 expression was L288 let us identify AMPs….
Round 2
Reviewer 1 Report
This article should be published in present form